# Exploring the Impact of Storytelling for Hospitalized Patients Recovering from COVID-19

**DOI:** 10.3390/healthcare11040589

**Published:** 2023-02-16

**Authors:** Lara Gurney, Vincci Chung, Maura MacPhee, Evelyn Chan, Claire Snyman, Jaclyn Robinson, Serena Bertoli-Haley, Elizabeth Baron

**Affiliations:** 1Vancouver Coastal Health, Vancouver, BC V6T 1Z4, Canada; 2School of Nursing, University of British Columbia, Vancouver, BC V6T 2B5, Canada

**Keywords:** patient experience, staff experience, patient stories, healthcare leadership, healthcare, patient-orientated research, workforce engagement, healthcare, COVID-19

## Abstract

There are mental and physical deficits associated with COVID-19 infection, particularly among individuals requiring hospitalization. Storytelling is a relational intervention that has been used to help patients make sense of their illness experiences and to share their experiences with others, including other patients, families and healthcare providers. Relational interventions strive to create positive, healing stories versus negative ones. In one urban acute care hospital, an initiative called the Patient Stories Project (PSP) uses storytelling as a relational intervention to promote patient healing, including the development of healthier relationships among themselves, with families and with healthcare providers. This qualitative study employed a series of interview questions that were collaboratively developed with patient partners and COVID-19 survivors. The questions asked consenting COVID-19 survivors about why they chose to tell their stories and to flesh out more about their recovery process. Thematic analyses of six participant interviews resulted in the identification of key themes along a COVID-19 recovery pathway. Patients’ stories revealed how survivors progress from being overwhelmed by their symptoms to making sense of what is happening to them, providing feedback to their care providers, feeling gratitude for care received, becoming aware of a new state of normal, regaining control of their lives, and ultimately discovering meaning and an important lesson behind their illness experience. Our study’s findings suggest that the PSP storytelling approach holds potential as a relational intervention to support COVID-19 survivors along a recovery journey. This study also adds knowledge about survivors beyond the first few months of recovery.

## 1. Introduction

### Background

Numerous physical and mental health sequelae have been associated with COVID-19 infection. In one scoping review, 23 studies focused on physical health status where the common problems were fatigue, joint pain and decreased exercise capacity. Mental health outcomes were evaluated in 18 studies where common mental health issues were anxiety, depression, post-traumatic stress disorder (PTSD) and cognition deficits. The review found that physical and mental problems were more pronounced in individuals who required hospitalization. Reviewers also commented that COVID-19 symptoms are similar to chronic fatigue syndrome, which is a bundle of physical and mental deficits associated with other viral infections [1]. A phone follow-up study completed mental health assessments on hospitalized COVID-19 patients at time of admission, within two weeks of admission and at one month after discharge [2]. These researchers established a one-month trajectory of recovery from anxiety, depression and PTSD, where 80% of patients showed improvement, but 20% of patients with longer hospitalizations had more severe, persistent mental health symptoms. The researchers surmised that, during COVID-19, hospitalized patients who were socially isolated for longer periods of time had more severe mental health problems [2]. 

Earlier studies of viral infection showed that chronic fatigue syndrome can persist for up to six months or more [3,4]. To date, most research on post-COVID-19 symptoms has examined the first few months of recovery, although researchers and practitioners are beginning to appreciate “the long-term emerging multi-system impacts of the disease…” [5] (p. 1). There are estimates that 10% of post-COVID individuals living in the community have persistent symptoms that adversely affect their capacity to care for themselves or to work and support themselves and their families [6]. 

A number of outpatient or community-based rehabilitation care models are emerging to assess, track and treat post-COVID-19 multiple systems involvement, although initial focus has been on the acute phase (i.e., first few months) of recovery from COVID-19 [7]. Rehabilitation or recovery interventions target the most prevalent symptoms of dyspnea, fatigue, musculoskeletal, cognitive and mental health impairments. A systematic review on rehabilitation interventions predominantly evaluated the capacity of physiotherapy and respiratory therapy to improve physical activity tolerance and pulmonary function. Results were mixed due to heterogeneous outcomes measures and small trial sizes [8]. Missing from the post-COVID-19 literature were reports of relational interventions to address mental health and social care needs [7,8,9]. 

Relational interventions are designed to address patients’ social connections and quality of social relationships among family, friends and the community [10]. Relational interventions have been effective in addressing common mental health problems and quality of life perceptions in patients with a variety of physical and mental health diagnoses, including brain injury recovery [11] and stroke survivorship [12]. In these latter instances, there can be persistent, long-term physical and mental health sequelae beyond the immediate acute phase of injury. 

One effective relational intervention is narrative therapy, or storytelling, with guided support by a trained facilitator or therapist [12]. Narrative therapy helps a patient construct a story of themselves that focuses on positive and valued experiences versus negative ones. Illness and trauma often create a sense of loss and failure in patients, and the narrative approach engages them in “re-authoring” their life stories to reflect a better, stronger and more capable self-identity [13,14,15]. 

Another approach to storytelling is the Ganz model of Public Narrative [16] that was developed for public engagement, action and advocacy. This model has been promoted through the Institute of Healthcare Improvement (IHI) as a way of leading change at different systems levels, beginning at the individual level with the “story of self.” At this level, self-reflection and raised awareness of core values and beliefs result in an individual’s construction of their own, meaningful story. Once an individual knows themselves better, they can share their values and beliefs with others to create a collective identity known as “story of us.” Collective identity is a pre-requisite to identifying common goals and actions, or the “story of now” [17]. A patient’s story is a powerful tool in the provision of holistic care, and stories are a fundamental way for people to make sense of the world and to learn and understand more about themselves and others [18]. Understanding a person’s experience through narrative or storytelling offers a glimpse into another’s life history, values, meaning, relationships, expectations and commitments [19]. Stories are reflective, creative and value-laden, usually revealing something important about the human condition [20]. Evidence suggests that the creation and sharing of personal stories by patients can empower patients to engage in their care, increase a sense of well-being and help others suffering from similar illnesses understand their own experiences, as well as letting the patient themselves feel less alone [20]. Furthermore, in the context of chronic disease self-management, Gucciardi found that storytelling encourages patients to take on a more active role in their health management and enables them to form strong bonds with others with similar health challenges [21]. 

Given the benefits of the storytelling approach, our healthcare organization adapted the Ganz Public Narrative model to create a means for the patient experience or patients’ stories to be shared within our broader community of patients, families, visitors and staff.

Our organization’s Patient Stories Project (PSP) originated in a single Canadian Intensive Care Unit (ICU) at a large, urban quaternary care hospital in 2016. The purpose of the PSP was to create a community of appreciation and understanding through patients’ individual stories. After a successful pilot in critical care and emergency services areas of the organization, the PSP became part of the patient experience portfolio, a division of quality and patient safety. A patient experience team, including patient partners, oversees the PSP and other patient experience initiatives, including evaluation of outcomes pertaining to Quadruple Aim patient and staff experience. The team utilizes multimedia to recruit patient participation and to share stories broadly via posters throughout the hospital and online [22].

Due to burnout and nursing shortage problems, the team began its PSP evaluation with a focus on patient stories’ impact on nurses. Team members collaborated with nurse researchers to conduct a qualitative study of the PSP. Thematic analysis of data from focus groups and interviews with critical care nurses found that the majority of patients’ stories included appreciation for the compassion and care of their healthcare providers, particularly nurses. Nurses described how PSP stories were very meaningful to them, especially patient recognition of the relational care they provide—above and beyond the physical and technical tasks of nursing. Patient storytelling through the PSP also gave nurses insight into the patient experience and made them more aware and accepting of patient needs and preferences [23].

In the spring of 2020, COVID-19 overwhelmed the health care system and burnout threatened the mental and emotional well-being of health care teams everywhere. The PSP was formally expanded to all the units caring for COVID-19 patients: the emergency department, ICU, medicine units and the COVID-19 outpatient recovery clinics. The purpose of the expansion was to build on successes in the pilot, where nurses, in particular, acknowledged the many positive benefits of PSP stories. Patients with COVID-19 were invited to submit their stories of hospital care by completing the PSP template and submitting them via email or mail. 

With the pandemic easing in 2021, the patient experience team began to consider the positive effects of storytelling for COVID-19 survivors. On the PSP consent form, participants can consent to follow-up by the patient experience team. A decision was made, therefore, to contact COVID-19 survivors in the community at least three months after discharge from the hospital to better understand their recovery trajectory. So much is unknown about this novel illness; we wanted to hear from COVID-19 survivors, in their own words, about their recovery experiences. We also postulated, based on benefits of storytelling in other chronic illness contexts, that PSP storytelling might therapeutically benefit those recovering from COVID-19. Our study premise was: the PSP holds potential as a relational intervention to support COVID-19 survivors in their recovery journey. 

The aim of this study was to understand the impact of patient storytelling from the perspective of patients who were hospitalized with COVID-19. A secondary aim was to learn more about patients’ recovery journey beyond the first few months of hospital discharge.

The sections of the paper are as follows. The Methods section provides a description of the qualitative design we used and our patient-oriented research approach. The Results section consists of seven sub-sections for each major theme that corresponds to patients’ experiences after the acute phase of COVID-19 recovery. There are exemplar quotes accompanying each theme, and the final Results sub-section is our proposed recovery pathway for COVID-19 survivors, based on the seven patient experiences themes. The Discussion section links our findings to relational intervention literature that includes storytelling approaches for patient therapy and healing. The Conclusions summarize unique contributions from this study while acknowledging limitations, such as the need for a larger and more diverse study sample. 

## 2. Materials and Methods

### 2.1. Study Design

In this study, we utilized qualitative methodology by employing the narrative approach to life stories method [24] to gain an in-depth understanding of the patient experience through storytelling. Thematic analysis was used to systematically identify and explore qualitative themes brought forth by study participants. We used Braun and Clarke’s six phases of thematic analysis, including member checking for rigor [25].

Our organization follows a national patient-oriented research approach where patient engagement and co-production are utilized in research [26,27]. Patient-oriented research aligns with patient-centered care, and the importance of ensuring patients’ expertise is integrated throughout the research process [27,28,29]. Patient partners and COVID-19 survivors were actively engaged in this study as members of our hospital’s patient experience team; they played an integral role in question development and piloting, interviews, data analysis, validation/rigor and writing. 

### 2.2. Ethical Approval

The study was approved by the institutional ethics review board of the hospital (V21-03946) and The University of British Columbia (UBC) (H21-03946). All subjects provided written and informed consent. 

### 2.3. Study Setting

This research was conducted in a province of Canada. The setting of the study was a 1000 plus bed, quaternary care, inner city teaching hospital, serving a mixed medical and surgical patient population. 

### 2.4. Sampling

Patients who participated in the hospital’s COVID-19 PSP were asked if they wished to be contacted for future research purposes. Patients who gave their informed consent and contact information were invited by phone and/or email by a patient experience team member to participate in this study. The inclusion criteria were English-speaking patients over 18 years of age who were hospitalized or followed in the post-COVID recovery unit for COVID-19 infection between April and October 2021 or between 6 months and 1 year from hospital discharge. An exclusion criterion was a patient’s cognitive inability to answer researcher questions.

### 2.5. Data Collection and Interviewing Questioning 

One-hour long interviews with patients were conducted using the Zoom platform between January and April 2022. The interviews were completed by members of the hospital patient experience team and UBC affiliates. See Appendix A for the interview guide questions. 

### 2.6. Analysis

Interviews were transcribed using an automatic text transcription software, TEMI. The research team followed Braun and Clarke’s analytic process [25]. At a descriptive coding level, four of the authors coded each of the transcripts for key words and phrases. A coding framework of descriptive labels and exemplar quotes was collaboratively developed and shared with other members of the team, including two patient partners. Through the data review and re-review process, mutually exclusive themes were created until all the data were captured.

## 3. Results

Five study participants required hospitalization for COVID-19, and one participant needed prolonged outpatient care in a COVID-19 recovery clinic. The six study participants had significant sequelae, such as severe fatigue, cognitive impairment and breathing difficulties/dyspnea, including signs and symptoms that continued or developed after acute COVID-19. Five volunteers were Caucasian and one volunteer was Southeast Asian. Four volunteers were between 35 and 55 years of age and two volunteers were between 70 and 80 years of age. Five volunteers identified as male and one volunteer identified as female. Appendix B displays the participant demographics and the number of months between discharge and when their interviews were conducted. Through our analysis of their accounts, we inductively identified seven themes that captured each patient’s experiences. The seven key themes are: not ready, making sense, providing feedback, and seeking closure, expressing gratitude, new normal, taking control and creating meaning. The following sub-sections provide descriptions of each theme with substantiating quotes where pauses have been removed. 

Not Ready to Share

This theme captures patients’ readiness to share their story. Study participants indicated there were cognitive, emotional, and physical roadblocks to story sharing, particularly due to COVID-19 aftereffects. 

“So it’s the cognitive abilities are way lower than, than we’re used to. And so it’s really hard to, to come forward and do, that’s why, you know, everyone’s like, you wanna do it on video. I’m like, I, at that point there was no way in the world. I couldn’t find my words. I was mixing up words all the time. So email was really the only option for me.” (Participant E)

One participant contemplated that some people may equate storytelling with reliving a traumatic experience, and therefore choose to avoid storytelling.

“It’s sometimes people [who avoid storytelling], because it’s such a traumatic experience for them. They don’t necessarily want to relive it or talk about it. So it’s easier to just shut off … I’m not sure how to phrase it, easier to people.” (Participant C)

Another participant reflected on being in an emotional place of readiness to share one’s story:

“I haven’t touched even like my story because it’s been so emotionally, like I lost my hair too. Yeah. I had hair down to here and that fell out Christmas and in clumps. So that was another emotional thing I had to go through.” (Participant F)

Making Sense

Study participants shared how they utilized various strategies to document and make sense of their COVID-19 illness experience and recovery journey. When asked about the merits of the PSP, documenting and sharing lived experience with others was a way for them to understand what was happening to them.

One study participant recorded their daily symptoms in a notebook, to track and recognize patterns between activity level and long COVID-19 symptoms. For this participant, understanding patterns helped inform their daily activity choices and increased their sense of control. 

“I track all my symptoms every day and so this is February [holds up notebook]. All the red marks are how bad those symptoms were on that day. So you can see it kind of comes in waves, but there’s still stretches where like barely any symptoms for like a two week stretch here where here it’s like, obviously that, that was a bad few days… but this is one out of out five. I don’t have last year’s book. But last year’s book, there are days where it’s just red across the board, every single symptom, you know, headaches all day, chest pains, all day dizzy nausea, exhausted, can barely breathe. And so looking back on that history also gives me that reflection and motivation too.” (Participant E)

Another study participant chronicled their near-death experience on their phone and texted their journal entries to a good friend. This experience became a topic of discussion and reflection between them. 

“…but that experience, I was able to write in detail what I experienced when I went out and it was pretty fantastic. I gotta say, if I went to the other side, it was very blissful. I wrote them in detail. I don’t know what happened, but here’s an experience I just had. And so I keep that because I wanna know what I said <laugh> so they [friend] have all my texts and writings about that, you know, I was able to write and chronicle after that, cuz I felt uplifted.” (Participant F)

Another participant shared their illness experience and made connections with others through social media. Although participants chose different ways to document, organize, and share data, they were all trying to process and comprehend what was happening, and/or what had happened to them. Sharing their illness experiences was an opportunity to capture the moment for future reflection, to process the event, and to weave the illness experience into their overall life story.

Providing Feedback and Seeking Closure

Study participants shared how the PSP gave them an opportunity to provide follow-up and feedback to their health care providers after discharge. They wanted to let those who cared for them know how they were doing. 

“One of the people I talked to was a pharmacist, very young man. And he was extremely good at what he did. And there was a couple of things we talked about doing in my case and we agreed, okay, we’ll do this. And so I go off and a week later I’m all fixed up because of what he did for me. Yeah. Does he ever hear back? No, I know. Yeah. So that, to me is a problem.” (Participant A)

“One is, you know, when I would talk to the therapist specifically at the post COVID clinic about my gratitude for the progress I made , I could sense in them a frustration because I feel like this is hard to explain when you’ve got long COVID. It’s really hard to think, and it’s really hard to follow directions and make changes to your life. And I could sense in them the frustration, because they’re telling all these patients what they can do to make their life better. And only a few of us I know, were, are following the program. And so I wanted them to know that their voice is being heard and they’re changing people’s lives.” (Participant E)

Participants hoped to bridge the informational and emotional gaps between their experience and their health care providers, with the aim of improving care for future patients. 

“I’d love to be able to have a message to say, Hey, what you told me on this date worked. I do it with my dog. My dog has a condition. And I take her to the vet and then the vet says, okay, let’s try this. Well, a week later I get the vet and I say, that worked. And he goes great. Now I know for next time. Yeah. Yeah. That feedback is really not available to us in the human side.” (Participant A)

“I decided to share my story because I found a lot of things that could have gone better for me in my care and at the hospital. They forgot to give me inhalers. I had COVID pneumonia. I learned later from another doctor that they had forgot to give me cortisone and a steroid and they lay me on my back and I should have been propped up just all these various things that I was, you know, pondering after I got out of this hurricane kind of thing. I was like, what happened? And then I’m like, okay, what could have gone better?” (Participant F)

Some participants thought a formal feedback process after discharge from an acute care setting should be a routine component of anyone’s journey through the healthcare system.

“There should be a normal process for somebody to be released and then to say, okay, what are your thoughts. Yeah. But more important is to say, how are you doing? Yeah. To say, we don’t just push people out the door and then assume their GP will take it from there. Maybe that’s what people are thinking. Maybe hand it over the GP. But no, to say to the person leaving the hospital, we’re gonna be watching and we’ll be, we’ll be checking on you. I think it would actually be a level of additional care that people could recognize.” (Participant A)

Expressing Gratitude

An additional theme was participants’ desire to express gratitude for the care and compassion shown by health care providers and other staff during their acute illness journey, from housekeeping staff to the food service workers, porters, security guards, and more. During their interviews, participants frequently described thankfulness for the care they received and their desire to “give back”. Their words were often accompanied by visible tears, voice changes and facial expressions. 

“…all these people that I tried to consider when I did the little story [before discharge], it my way of, of thanking them. When they [patient experience team] said they were gonna put it on the wall, I thought, well, geez, that’d be great if I could actually mention these. And the people I’m mentioning-- they’ll know who they are. They’ll recognize themselves by what I’m saying about them. Or what I said about one person, two or three people might take that as, oh, that that must be me. Well good, because I’m sure they were helping other people and those other people maybe didn’t take the time to try to thank them. So that was just my way of trying to give something back.” (Participant B)

Participants acknowledged that health care workers are also only human and are just as afraid of the unfolding COVID-19 situation as everyone else. Participants felt grateful for health care workers’ bravery and willingness to take risks on their behalf.

“I think people seem to forget that healthcare workers are human beings too, and fear is a normal response to the unknown. So just like we are scared of it, you guys are just as scared of it and you see more of the outcomes, specifically, the bad ones that we do not. And I think we should be more appreciative of the risks that you take to protect us.” (Participant C)

Participants were deeply appreciative, admitting that health care teams had given them the most precious gift: their lives.

“So, you know what, I guess the most impactful thing for me is to realize that, I mean, I owe my life to people’s efforts along those lines. To people who basically, as heroes go are unsung.” (Participant A)

New Normal

The act of storytelling produced a ‘new normal’ for patients. When asked about life after COVID-19, participants verbalized becoming aware of and acknowledging that they were at a new level of physical and/or cognitive function. One participant commented on their struggle to accept how they were functioning with less clarity than before their illness. 

“It’s really hard because I know I received the information before I started acting on the information and that level of cognitive clarity for somebody who’s based my whole career on my brain, you know, everything I’ve done has been cognitive. And so to think that I was getting information and not acting on it is, was really hard for me to swallow. And that’s the thing. I think caregivers have a hard time understanding who they’re talking to because the patient is different.” (Participant E)

In turn, family, caregivers and other health care providers working with the individual also had to develop awareness and acknowledgement of the individual being a different person post COVID-19. This required an adjustment to their care approaches and techniques. 

“You know, I remember reading over my doctor’s notes about the long term disability application, you know, like [participant’s name] is very amicable and friendly and dah da da, da, da. And I was like, she’s still talking about me as if that’s who I am and it’s not who I am anymore. She (doctor) is gonna have to tell me things two or three times before I listen and I don’t have that same ability anymore, but she hadn’t adjusted her care for my current abilities yet. And her perspective of me--- I just think it’s so hard for people to understand until they’ve been there.” (Participant E)

When faced with challenges in daily life related to limitations in one’s physical or cognitive abilities, participants voiced determination to keep trying.

“I do that more often than I would like to right now I’ll search for words. Uh, the words are there. It’s like I’ve opened a filing cabinet drawer and I’m looking for the file to pick and I don’t get it first grab, so I keep grabbing and I do get it, cuz it’s still in there.” (Participant B)

A participant described having to make trade-offs, such as having to stop working and to go on long term disability, in order to focus on getting better.

“I’ve really had to swallow a lot of my pride to [admit I’m not] capable of working… to recognize that it’s okay to say you can’t work. And a lot of people I see are still suffering. They haven’t taken time off work. They’re still in full time jobs and their recovery. And I could have never done this if I was working, you know, never in a million years, I wouldn’t have made it. It’s okay to say I can’t work for six months or a year while I get better from this illness that’s shut down the planet.” (Participant E)

Another participant described the reality of discovering that they are no longer who they used to be. 

“Basically how it overtakes you, it’s a whirlwind storm that you have to weather and then, once you get out of it, everything is in pieces, you know, and you can’t just go back to how you were before because she doesn’t exist anymore. There’re changes. Right?” (Participant F)

Taking Control

‘Taking control’ describes life after COVID-19 and participants’ description of regaining control of their lives after accepting and adapting to their new normal. At the time of interview, two of our participants had already returned to their prior level of function, but the rest were still living with differing symptoms of variable severity. Regaining control of their lives varied, depending on each participants’ condition. Some participants described regaining control of their lives by gradually working their way back to their prior level of function, while others took back control by accepting changes in themselves and finding strategies to tailor their daily routines so that they could carry on. 

“When I came back [from the hospital] I dropped from my normal 220 pounds. I was down to 170. Wow. And I needed a walker to get from my home office here to the kitchen. And, I had therapy for about three weeks, but it was just steady…like, okay, today I can go 20 feet. And it was after a week that I was able to get down the back steps and leave my yard again. Yeah. So each little increment of being able to walk across the street. And then eventually it was not for eight weeks that I actually did get back on a bike. I had to wait for my various incisions to heal up, but, [it was] a steady climb back to where I was before.” (Participant A)

Other participants shared acceptance and strategies to adapt to realities they faced in their new normal. 

“But then eventually, you know, the physiotherapist she’s like [participant’s name] buy, a shower stool. I’m like fine, I’ll do it. It was so good. Like I get my stool and put it in the shower and I can enjoy long showers again. And you know, I have to have it on cold for part of the shower because otherwise that’s still super exhausting, but you know, I can do these things that I enjoy.” (Participant E)

Although they were not who they used to be, participants came to accept who they are now.

“I’m still one of the lucky ones. I know I’m one of the lucky ones that I, I was nine days in hospital. And I got to walk out and I would say I’m 95% of what I used to be, and that’s not bad for the seriousness of this disease.” (Participant B)

Creating Meaning

‘Creating meaning’ illustrates how participants found deeper meaning behind their illness experience. All study participants underwent the negative experience of being ill with COVID-19, and everyone was able to identify an important lesson after their illness experience. Experiencing a serious illness usually involved some degree of loss of control that required reflection and creation of an explanation for what happened to them. In many instances, participants felt that sharing their experiences with others was one way to create meaning for themselves. 

One participant reflected on the seriousness of COVID-19 and wanted to let others know that prevention is better than treatment.

“People kind of know that it is serious. There are effects, but at the same time, if you are careful, if you take the precautions where you mask and social distance. If you’re sick or if you feel sick, stay home. These are all things that can lighten the burden for everyone around us. And it’s not as if, yeah, I’m just going to get a little flu and then it goes away-- for some people there are long-lasting effects and it doesn’t go away overnight. It stays, it lingers and it affects every aspect of their life. I’m constantly tired because my lungs are just not capable of breathing again, and I get winded really quickly. So if you can prevent it, it’s so much better.” (Participant C)

Another participant described himself as a messenger, trying to deliver an important announcement regarding vaccination. 

“Sharing it, like fine. I’ll be God’s messenger. And I’ll be, you know, spreading this love around and you know, like whoever you are, like, I care for you and this is my experience do with it, what you will. Right. I encourage you to go this way, but I can’t force you.” (Participant D)

Participants described positive feelings and sense of purpose from knowing that they were able to help others. Participants’ active efforts to educate, inform, and/or help others provided them with meaning and purpose through goal-driven interactions. 

“It makes me feel really good when I know that I’ve made a difference for people, you know, like just yesterday, helping somebody out with smell therapy, you know, like I know that she’s gonna get better because of me helping her and you know, I love it.” (Participant E)

“I mean, you know, who knows if my story can touch someone reading because everyone has challenges and everyone gets ill at times in their life and it doesn’t necessarily have to be COVID too, you know.” (Participant F)

A Thematic Pathway

Through identification of key themes, a pathway emerged where post-COVID-19 patients progressed from being overwhelmed by their symptoms to making sense of what was happening to them, providing feedback to their care providers, feeling gratitude for care received, becoming aware of a new state of normal, regaining control of their lives, and ultimately discovering meaning and an important lesson behind one’s illness experience. 

## 4. Discussion

Based on our study findings, we believe that storytelling is a relational intervention that can help a post-COVID-19 patient proceed along their recovery journey. To our knowledge, this is the first qualitative study to map out potential recovery beyond the acute phases of the viral infection. 

Storytelling can establish causal links between actions and events [30]. Storytelling allows others to build, use, and update their knowledge, becoming better prepared to respond to unexpected situations [31]. Stories make it easier to understand complex or foreign concepts. Our proposed causal pathway of recovery may help promote understanding among other COVID-19 survivors, providers and the public, with respect to what happens after serious COVID-19 infection and hospitalization. For our participants, storytelling through the PSP was an opportunity to let others learn from their experience without having to endure the pain of undergoing the same experience. 

Lipsey et al., 2020 found that patient storytelling has potential to educate, motivate, and reduce fears for others who are in a similar situation. For listeners and learners, it is easier to understand a story from a person who has faced a similar situation versus didactic education [18]. This is in line with our participants’ observations that information they provided as a person with lived experience was more believable and credible than information given by experts. Storytelling from previous patients, who have faced or overcome real-life health challenges, can be an effective way to engage patients and influence their decisions and actions which impact health [18].

Stories make events emotionally meaningful, allowing us to better understand and accept them [32]. For significant others, their loved ones’ stories help them understand the impact of the experience and to fully appreciate the need for comfort and support during a challenging time [33]. When patients survive and even thrive after crisis, their stories of successes and challenges can facilitate a sense of cohesion and meaning making for current and future generations [34]. People who have experienced a crisis or a major life changing event often feel the need to share their story and give voice to their experience [34]. One’s sense of personal identity relies on their ability to tell a coherent narrative of the past, present, and future [34]. As Figure 1 illustrates, participants’ narratives linked together themes for a hopeful way forward from COVID-19 losses. As described in Ganz’s Public Narrative model of storytelling, “stories of self” evolve to become “stories of us” and “stories of now” [35]. 

Similarities exist between our COVID-19 recovery pathway and contemporary grief theory. Therapists have generated evidence-based strategies for healthy grieving in the aftermath of COVID-19 [36,37]. Meaning-making is one beneficial approach for adaptive coping with grief. Grief theory postulates that recovery is not necessarily linear, and there can be points of regression when extra supports are needed. Similarly, ongoing recovery from COVID-19 may have stalls, dips and reversals. For example, when we asked our participants why some people do not want to tell their stories, a common response was about readiness to share a personal and painful experience, often an experience of loss. When patient storytellers retell experiences that were emotionally and mentally difficult for them to manage, there is the risk of re-traumatization [38,39]. Patients, therefore, need to determine when and what to share according to their personal readiness. Follow-up with COVID survivors and opportunities for check-ins at regular intervals may be one way to ensure support is available during the return to a ‘new normal.’

From the provider perspective, storytelling is a way to better connect to patients’ lived experiences, to learn from them, and to enhance their recovery experience over time [40]. Storytelling holds potential as a means of informing direct care providers and organizational leadership of what matters most to individuals faced with many life challenges and unknowns. Our healthcare organization’s support for the PSP complements organizational commitment to the IHI’s Quadruple Aim mission to enhance the experience of care for individuals, as well as staff experiences during care delivery [41]. More evaluative work remains to demonstrate other links between the PSP and the Quadruple Aim goals of better population health and cost efficiency; however, the premise of Quadruple Aim is that enhancements through one aim will synergize the others [41].

Patient narratives and patient experience data have been used in hospital quality improvement campaigns [42,43,44]. A scoping review found that patient experience data for quality improvement purposes are often collected via surveys and patient complaints and risk management [42]. As stated by the reviewers, “To facilitate knowledge exchange and promote the uptake of quality improvement initiatives…it is critical to understand the context in which initiatives have positive or negative outcomes” [42] (p. 178). The PSP approach is a unique, novel way of encouraging patient engagement by highlighting service users’ perspectives of what works well and what needs work. The PSP can provide contextual richness in patients’ own words. The PSP approach to understanding patient experience also provides concrete, explicit examples through the patient lens of how care can be more patient-centered. A US study compared content in patient narratives to patient experience survey data. Unlike survey data that focused on missed care or potential/actual errors, the narrative data focused on relational aspects of care, sharing information and examples that was not captured by existing surveys. In addition, three-quarters of patient narratives included carefully considered actionable items to improve healthcare delivery [44]. 

This study has added to our appreciation of how patient storytelling can be a relational intervention for COVID-19 survivors. In addition, this study has added new knowledge of what happens to COVID-19 survivors beyond the first months of acute recovery from viral infection. Other research suggests that a PSP approach to storytelling and sharing may generate a ripple effect with benefits to other patients and families, care providers and the organization [45]. Patient experience requires a culture shift where healthcare organizations recognize that quality relationships, empathy, understanding, and respect are desired by patients, families, and healthcare providers alike [45]. Through storytelling, we can move the pendulum away from the transactional business of healthcare towards a more relational focus [23].

## 5. Conclusions

Storytelling projects, such as the PSP, may be a therapeutic way to support patient recovery after a serious illness, such as COVID-19 recovery. The stories of COVID-19 survivors revealed themes along a pathway that extended beyond the first few months of recovery from viral infection, particularly when COVID-19 survivors are back in the community with expectations of returning to life as usual. Our COVID-19 pathway can be a guidepost for patients, their families and survivors who are trying to make sense of what happens after discharge from hospital with COVID-19 physical and mental health sequelae. Further study is recommended to include a larger sample of participants having different ages, ethnicity, gender and body mass index.

### Limitations

Limitations of this study include being set within one hospital only with a small sample of six participants. There was limited participant representation with most participants being Caucasian male. 

## Figures and Tables

**Figure 1 healthcare-11-00589-f001:**
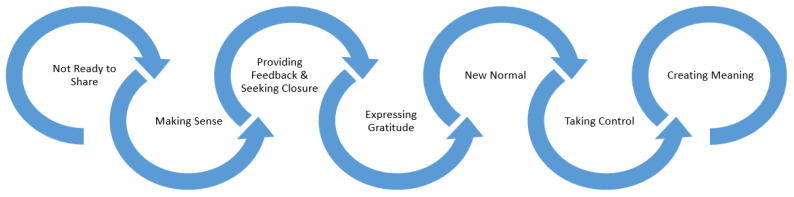
Pathway through COVID-19 Recovery. Participants progressed through the above steps during their COVID-19 recovery journey.

## Data Availability

Not applicable.

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
