# Peer review of "Exploring the Impact of Storytelling for Hospitalized Patients Recovering from COVID-19"

_healthcare, 2023, doi:10.3390/healthcare11040589_

Round 1

Reviewer 1 Report

The article entitled  The Patient Stories Project: Exploring the Storyteller’s Experience is a document of interesting subject matter.

The idea of the work is good and the topic is important. However, the reported work  requires improvement and revisions.
Your abstract should clearly state the essence of the problem you are addressing, what you did and what you found and recommend. That will help a prospective reader of the abstract to decide if they wish to read the entire article.
paper must provide a comprehensive critical review of recent developments in a specific area or theme. It is expected to have an extensive literature review followed by an in-depth and critical analysis of the state of the art, and identify challenges for future research.

The authors should do the analysis the conclusion section must clearly establish a strong correlation with the proposed topic.
The conclusion section can be refined better. Please indicates if your objectives were reached, in what your work is novel and confirms or not, previous findings. Also, ‎‎some perspectives generally arise from your investigations and must be indicated here. ‎‎

The interpretation of the experimental results should be significantly improved. In other words, pay attention on the interpretation of the experimental results. Only their presentation is not enough for a scientific paper.

The abstract and conclusion parts must be more informative by including more mathematical findings.

Author Response

Dear Reviewer,

Reviewer Response Report

Reviewer

Round

Report

Response

1

1

Your abstract should clearly state the essence of the problem you are addressing, what you did and what you found and recommend. That will help a prospective reader of the abstract to decide if they wish to read the entire article.

We have revised the abstract to provide background, aim, methods and outcomes.

1

1

paper must provide a comprehensive critical review of recent developments in a specific area or theme. It is expected to have an extensive literature review followed by an in-depth and critical analysis of the state of the art, and identify challenges for future research.

We have added literature background on patients hospitalized for COVID-19, different types of rehabilitation models and interventions, our focus on relational interventions, and a description of the Public Narrative model we used for our study.

1

1

The authors should do the analysis the conclusion section must clearly establish a strong correlation with the proposed topic.

We have summarized key findings in the Discussion and Conclusion sections that link to our study aims.

1

1

The conclusion section can be refined better. Please indicates if ‎your objectives were ‎reached, in what your work is novel and confirms or not, previous findings. Also, ‎‎some perspectives generally arise from your investigations and must be indicated here. 

We have highlighted our unique findings and we have presented more literature on uses for storytelling at different systems levels. We have also made clearer links to Quadruple Aim, particularly the importance of patient experience (one of the 4 aims).

1

1

The interpretation of the experimental results should be significantly improved. In other words, pay attention on the interpretation of the experimental results. Only their presentation is not enough for a scientific paper.

We have tied the themes and construction of the recovery pathway to other evidence of recovery from COVID-19. This was a qualitative study and the quotes we’ve included in Findings are  data exemplars. We took out pauses and tightened up quotes to be clearer to readers.

1

1

The abstract and conclusion parts must be more informative by including more mathematical findings.

We did not do surveys or generate quantitative data. We did add demographics about our six participants.

For more details please see the revised version manuscript.

Reviewer 2 Report

Dear Authors,

The manuscript title: The Patient Stories Project: Exploring the Storyteller’s Experience.

My decision is to reject the paper as I can not see any novelty of the work conducted in the current format of the paper. It is likely an experiment with simple data collection methodology; even the questions used in Appendix A are very simple and doesn’t allow any kind of analysis to be conducted. There is also a major issue in the way of writing the paper as I cannot see any theoretical frame work or hypothesis to be tested out of these interviews. In addition, the literature review conducted is not sufficient.

I understand that the paper is about the story telling but still authors should identify main aims and objectives as well as research gap to be filled out of the story telling interviews. Finally, to be fair the idea of story telling is great however it needs to be design well to fill a research gap and to answer research questions and then authors can formulate this into a good Journal paper.

Author Response

Dear Reviewer,

Reviewer Response Report

Reviewer

Round

Report

Response

2

1

My decision is to reject the paper as I can not see any novelty of the work conducted in the current format of the paper. It is likely an experiment with simple data collection methodology; even the questions used in Appendix A are very simple and doesn’t allow any kind of analysis to be conducted. There is also a major issue in the way of writing the paper as I cannot see any theoretical frame work or hypothesis to be tested out of these interviews. In addition, the literature review conducted is not sufficient.

I understand that the paper is about the story telling but still authors should identify main aims and objectives as well as research gap to be filled out of the story telling interviews. Finally, to be fair the idea of story telling is great however it needs to be design well to fill a research gap and to answer research questions and then authors can formulate this into a good Journal paper.

This study was a patient-oriented research project, and patient partners and COVID-19 survivors were an integral part of our research team. They co-developed and piloted the research questions. They also helped with interviews, data analysis and interpretation.

We have included more description of relational interventions and the storytelling approach we used.

For more details please see the revised version manuscript.

Reviewer 3 Report

This article wants to understand the role of Patient Story Project in the  evolution of patients who were diagnosed with COVID 19.

Here you can find some suggestions for your manuscript.

  1. Title

I would recommend the authors to include the COVID 19 in the title so the writer can understand which is the focus population.

  1. Abstract

I would recommend a better structuration for the abstract. Maybe the first concept to explain is the Patient Story Project. And then I didn’t find a little explanation about the methodology, results or the final conclusions.

  1. Introduction

Line 50: why do you talk about the results in the introduction? This is confusing for me..

You only use 5 references in all your introduction, I think you could support it with more literature.

Do you have an hypothesis? Which is your research question?

  1. Methods

Line 80: I am not sure if this explanation should be in this section, it sounds more an opinion to me

Line 95: more information about inclusion and exclusion criteria should be added

Line 107: more information about the kind of software you used would be interesting for the readers

Line 109: which authors did every task ? the analysis section needs to be more detailed

  1. Results

I miss some sociodemographical information about the patients, some “quantitative” data before starting with the qualitative results…we don’t know anything about the six patients: gender, age, etc…

  1. Discussion

The concept of Patient Story Project is  also in the discussion but you didn’t talk about it in the methodology or in the results, it is not clear for me what is it and how you followed it through the intervention…

Which are the differences between what you call “storytelling” and normal qualitative interviews? These concepts are not clear for me after reading the whole manuscript..

Line 407: Institute for Healthcare Improvement is another concept that you haven’t explained before, it is strange to read it in the discussion

I recommend the authors to improve the structure and expression of the discussion, it could be redacted in a more clear way. I can’t see the discussion about the diferent sections of the results…

Line 415: I would not use the word “must”, as your sample is very small and your manuscript has limitations.

I can’t see a coherence between the introduction, the results and the discussion section, please reconsider it for a better understanding of the readers.

Thank you for taking into consideration these suggestions, best regards.

Author Response

Dear Reviewer,

Reviewer Response Report

Reviewer

Round

Report

Response

3

1

I would recommend the authors to include the COVID 19 in the title so the writer can understand which is the focus population.

Updated to:

Exploring Patients’ Experiences with COVID-19 who participated in the Patient Stories Project

3

1

 would recommend a better structuration for the abstract. Maybe the first concept to explain is the Patient Story Project. And then I didn’t find a little explanation about the methodology, results or the final conclusions

We have restructured the abstract to include more comprehensive information.

3

1

Line 50: why do you talk about the results in the introduction? This is confusing for me..

We have removed results from this section. We have updated line 112-120 to reflect.

3

1

You only use 5 references in all your introduction, I think you could support it with more literature

We have rewritten the background to include more literature on concepts relevant to this study.

3

1

Do you have an hypothesis? Which is your research question?

We had two aims versus research questions. Because this was an exploratory qualitative design, we had broad goals to better understand the impact of storytelling on patients, and to learn more about their recovery journey.

3

1

Line 80: I am not sure if this explanation should be in this section, it sounds more an opinion to me

We have changed our language to indicate testable ideas versus concrete evidence

3

1

Line 95: more information about inclusion and exclusion criteria should be added

We have added information about inclusion and exclusion criteria.

3

1

Line 107: more information about the kind of software you used would be interesting for the readers

Added at line 196.

3

1

Line 109: which authors did every task ? the analysis section needs to be more detailed

We have identified who was involved with different components of the study

Contribution listed lines 551-555

3

1

I miss some sociodemographical information about the patients, some “quantitative” data before starting with the qualitative results…we don’t know anything about the six patients: gender, age, etc…

We added a table with demographic information

Appendix B line 585

3

1

Which are the differences between what you call “storytelling” and normal qualitative interviews? These concepts are not clear for me after reading the whole manuscript..

We conducted qualitative interviews to find out the impact of storytelling on patients. In our organization, we are using storytelling as a relational intervention. The interviews were conducted to find out if this was the case or not.

3

1

Line 407: Institute for Healthcare Improvement is another concept that you haven’t explained before, it is strange to read it in the discussion

We have added in earlier information about the IHI and Quadruple Aim. IHI and the 4 aims are both important to healthcare organizations because positive patient experience is a major aim.

3

1

I recommend the authors to improve the structure and expression of the discussion, it could be redacted in a more clear way. I can’t see the discussion about the different sections of the results…

We have rewritten sections of the Discussion to have a more systematic flow, based on our Findings and their importance to patient experience for COVID survivors.

3

1

Line 415: I would not use the word “must”, as your sample is very small and your manuscript has limitations.

Deleted and used ‘encouraged’

3

1

I can’t see a coherence between the introduction, the results and the discussion section, please reconsider it for a better understanding of the readers.

We have made a stronger link across the sections of the paper. The Introduction and sections of the Discussion were rewritten to emphasize the importance of the themes (Results) with respect to understanding how stories helped raise awareness of factors important to successful recovery from COVID-19 infection and sequelae.

For more details please see the revised version manuscript.

Round 2

Reviewer 1 Report

Accept

Author Response

No revisions required. 

Reviewer 2 Report

Dear Authors:

Thanks for your efforts to improved the paper. Please find the following comments to improve the paper:

1.      Add at the end of your conclusion plan for future work in which this plan should include bigger sample of participants having different ages, ethnicity, gender and body mass index.

2.      Please consider to use square practice for the references [] instead of ().

3.      Add paragraph at the end of the introduction about the structure and sections of the paper.

4.      In page 4 replace the link by a number as you do not have to put the link in citation.

5.      Check the references at the end and make sure that all of them are consistent and put all fields.

All the best

Author Response

Revisions Second Round

Reviewer

Round

Reviewer

Answer

2

2

Add at the end of your conclusion plan for future work in which this plan should include bigger sample of participants having different ages, ethnicity, gender and body mass index

Added line 555-556.

2

2

Please consider to use square practice for the references [] instead of ().

Updated.

2

2

Add paragraph at the end of the introduction about the structure and sections of the paper

Added lines 152-162.

2

2

Page 4 replace the link by a number as you do not have to put the link in citation

Updated and removed. Line 168.

2

2

Check the references at the end and make sure that all of them are consistent and put all fields.

Double checked.

Reviewer 3 Report

Congratulations for your efforts in improving the manuscript, I don't have further suggestions for you, best regards

Author Response

No revisions required.